# Route to diagnosis of degenerative cervical myelopathy in a UK healthcare system: a retrospective cohort study

Bryn Hilton,[1] Jennifer Tempest-Mitchell,[1] Benjamin Davies,[2] Mark Kotter[2,3]

[1]School of Clinical Medicine, University of Cambridge, Cambridge, UK
[2]Department of Clinical Neurosurgery, University of Cambridge, Cambridge, UK
[3]Department of Clinical Neurosciences, Ann McLaren Laboratory of Regenerative Medicine, Cambridge, UK

**Correspondence to**
Dr Mark Kotter;
mrk25@cam.ac.uk

## ABSTRACT

**Objectives** Degenerative cervical myelopathy (DCM) presents insidiously, making initial diagnosis challenging. Surgery has been shown to prevent further disability but existing spinal cord damage may be permanent. Delays in surgery lead to increased disability and reduced postoperative improvements. Therefore, rapid surgical assessment is key to improving patient outcomes. Unfortunately, diagnosis of DCM in primary care is often delayed. This study aimed to characterise patients with DCM route to diagnosis and surgical assessment as well as to plot disease progression over time.

**Design** Retrospective, observational cohort study.

**Setting** Single, tertiary centre using additional clinical records from primary and secondary care centres.

**Participants** One year of cervical MRI scans conducted at a tertiary neurosciences centre (n=1123) were screened for cervical cord compression, a corresponding clinical diagnosis of myelopathy and sufficient clinical documentation to plot a route to diagnosis (n=43).

**Primary outcome measures** Time to diagnosis from symptom onset, route to diagnosis and disease progression were the primary outcome measures in this study. Disease severity was approximated using a prospectively validated method for inferring modified Japanese Orthopaedic Association (i-mJOA) functional scoring from clinical documentation.

**Results** Patients received a referral to secondary care 6.4±7.7 months after symptom onset. Cervical MRI scanning and neurosurgical review occurred 12.5±13.0 and 15.8±13.5 months after symptom onset, respectively. i-mJOA was 16.0±1.7 at primary care assessment and 14.8±2.5 at surgical assessment. 61.0% of patients were offered operations. For those who received surgery, time between onset and surgery was 22.1±13.2 months.

**Conclusions** Route to surgical assessment was heterogeneous and lengthy. Some patients deteriorated during this period. This study highlights the need for a streamlined pathway by which patients with cervical cord compression can receive timely assessment and treatment by a specialist. This would improve outcomes for patients using existing treatments.

## INTRODUCTION

Degenerative cervical myelopathy (DCM) is an umbrella term for compression of the cervical spinal cord due to degenerative

### Strengths and limitations of this study

► This is the largest study to characterise patient with degenerative cervical myelopathy flow and the first to do so within the UK healthcare system.
► Centralised, electronic case notes permitted accurate construction of patient flow pathways throughout the healthcare system.
► Sparse clinical records lead to the exclusion of some cases that would otherwise have been suitable and thus decreased sample size.
► This study was conducted in a single centre, and thus, further work is required to generalise conclusions.
► True modified Japanese Orthopaedic Association (mJOA) scores were not available, and thus, inferring mJOA was developed to retrospectively assess disease severity from case notes.

pathologies, such as osteophyte formation, intervertebral disc protrusion and ligament hypertrophy or ossification.[1] The estimated minimum incidence and prevalence of DCMs are 41 and 605 per 1 000 000, respectively, in North America.[1] However, this is likely to be an underestimate as DCM often goes undiagnosed and epidemiology frequently relies on operative incidence.[1] In a recent study of 183 randomly selected adults aged 40–80, 59% had radiological evidence of cervical cord compression and 1% as yet undiagnosed DCM. The evolution of incidental cord compression is unclear, but in the only study of its kind, Bednarik et al demonstrated 23% went onto develop DCM in a series of 199 patients.[2] These findings would suggest that DCM is far more common than currently demonstrated.

DCM causes progressive neurological dysfunction leading to significant disability and reduction in quality of life for patients.[3] Symptoms include limb spasticity, numb and clumsy hands, sphincter dysfunction, neck and limb pain, imbalance, and limb weakness leading to falls.[4] These symptoms often arise and progress insidiously in either

a stepwise or continuous manner.[5] The rate of progression varies between individuals and cannot currently be predicted.

A diagnosis of DCM requires congruent symptoms, clinical and MRI findings, typically evidence of spinal cord compression.[6] Surgery is the only evidence-based treatment for DCM and has been shown to halt disease progression and afford some improvement across a number of domains.[5 7 8] However, few patients make a complete recovery. This is due to the limited regenerative capacity of the spinal cord; as a result existing damage is often permanent and leads to lifelong disabilities. A recent study has demonstrated that patients with DCM have among the lowest 36-Item Short Form Survey (SF-36) scores of chronic diseases.[9] Therefore, improving functional recovery in DCM is a major unmet clinical need.

Recent international guidelines advise that all patients with DCM should be assessed by a specialist service and surgery offered for moderate to severe disease, as well as any progressive disease.[10]

Prognostic factors for postoperative functional improvement have been extensively studied. The main predictor of outcome is presurgical morbidity. Detecting progressive disease and early intervention is therefore important. Moreover, longer preoperative symptom duration is associated with poorer postoperative improvement.[11–16] In the largest cohort study of patients with DCM, surgery within 6 months of symptom onset offered the best chance of recovery. In this context, rapid referral to a spinal surgeon followed by prompt surgical intervention when appropriate is key to minimising disability.

Unfortunately, patients with DCM report significant delays in diagnosis.[17] This is corroborated in the study of Behrbalk et al[18] who found that most patients in Israel wait more than 2 years for a diagnosis.[18] If this could be overcome, outcomes would improve based on existing treatment strategies. While the delay in diagnosis is becoming better recognised, where and why the delays occur remain unclear.

Healthcare system research aims to characterise patient pathways, to identify the 'where' and 'why' of delays, to allow targeted logistical interventions. This process has been successful in cancer presentation and management,[19–22] informing clinician-based and public policy-based initiatives to improve cancer care.

Our objectives, therefore, were to characterise the patient with DCM pathways within the UK healthcare system in order to support targeted interventions aimed at accelerating diagnosis and management. This is the first study of its kind in the field of DCM.

## METHODS
### Study description
Retrospective, observational cohort study.

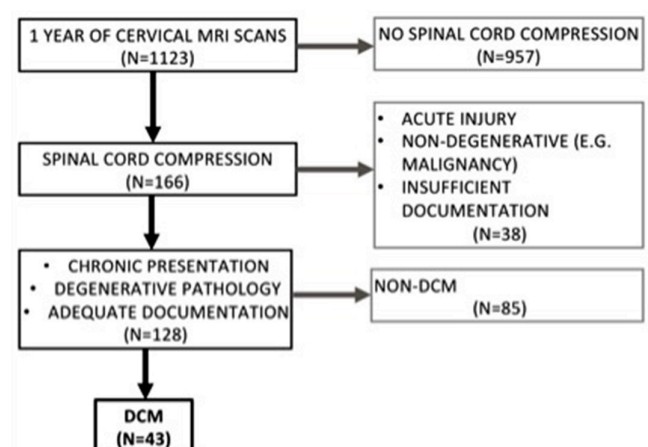

**Figure 1** Flow diagram depicting cohort formation methodology. DCM, degenerative cervical myelopathy.

### Cohort description
The radiological reports of all cervical spine MRI scans conducted over 1 year in a single, tertiary neurosciences centre were screened for descriptions of cervical spinal cord compression. The centre serves an estimated population of 5.9 million, although a considerable proportion of spinal patients are initially seen and are imaged, and may be managed in secondary care centres.[22] The clinical records corresponding to these scans were examined and cross-matched against inclusion criteria: non-traumatic cord compression due to spinal degeneration, a clinical diagnosis of DCM and adequate clinical documentation to characterise the referral pathway. Cases that met these criteria formed the cohort for this study (figure 1).

Peripheral hospitals in our region use different electronic and paper record keeping systems from one another and from this tertiary centre. Therefore, this study only included patients imaged in our centre, as this permitted full access to clinical documentation required to accurately document patient pathways.

### Patient and public involvement
This study was developed in response to patient reports of frequent and long delays in their diagnosis and management. These reports came from the community of myelopathy.org, an international DCM charity and were reconfirmed as a priority research area during a patient and public involvement day at the University of Cambridge. Some of these interviews were featured in a video documentary for Cambridge TV. While patients drove the theme for the study through their testimonials, they were not involved in the exact design of the study, which was informed by local referral pathways and literature on DCM. Once published, this study will be shared through myelopathy.org to make its findings widely available to patients with DCM and carers.

### Data collection
All relevant clinical records (assessments, correspondences, inpatient documentation) were examined for the cohort cases. Demographic data were collected: age

at time of onset, sex, relevant commodities, previous episodes of DCM and previous cervical surgery.

## Retrospective assessment of disease severity

The modified Japanese Orthopaedic Association (mJOA) is a common and well-validated assessment of disability in DCM.[10 23] A retrospective method for assessment of the patient's mJOA score was developed (table 1) and prospectively validated (n=13) using a kappa statistic. This was termed the inferred mJOA (i-mJOA). i-mJOA was developed as a research tool for this study to allow retrospective assessment of change in DCM disability, as grading tools are not routinely employed in clinical consultations. It was not specifically developed for wider usage, which would require thorough external validation prior to application.

### i-mJOA development methodology

1. A translation of mJOA into i-mJOA was designed based on commonly reported clinical details in patients' electronic records (eg, 'walking aid' vs 'no walking aid' imbalance, or 0–5/5 power grading). This translation is presented in table 1.
2. The electronic records of 13 patients assessed by a spinal clinic in our centre were collected. These were patients outside of the study range dates and thus were not included in our study. mJOA assessments were made during their clinical consultations, and therefore, a formal mJOA score was available.
3. These records were anonymised and the mJOA questionnaire results were removed.
4. The mJOA to i-mJOA translation was applied to these patient records.
5. True mJOA and i-mJOA scores were then statistically analysed using the kappa statistic. The results are presented below.

The i-mJOA and mJOA had good agreement (0.73), specifically upper limb (0.66), lower limb (0.75), sensory (0.32), sphincter (0.29) and total mJOA (0.73). Average differences between i-mJOA and mJOA ±SD were: upper limb (0.13±0.92), lower limb (0.13±0.92), sensory (−0.4±0.74), sphincter (0.27±0.70) and total mJOA (0.13±1.51).

This experimental tool was applied to assess the disease severity of our cohort. If no features for a subcategory of i-mJOA were documented, the category was assigned the highest score. i-mJOA subcategory scores were assumed to stay constant over time unless specifically mentioned to have improved or deteriorated. If the criteria were met for more than one score within a subcategory, the lower score was assigned. i-mJOA scores are reported as mean±SD.

## Analysis

Statistical analysis was carried out using SPSS V.22. Bivariate analysis was conducted between variables using Pearson's and Spearman's correlations depending on the variable types involved. Wilcoxon signed-rank test was used to calculate significant changes in i-mJOA from primary care referral to surgical assessment. As cases were selected on the basis of having sufficient documentation, there were no missing data.

## RESULTS

### Study demographics

A total of 1123 total scans were screened, resulting in a final cohort of 43 DCM cases (figure 1). The vast majority of these scans were of unique patients, although rarely patients were reimaged. The average age at time of symptom onset was 61.4±13.9 years and the majority of patients (28, 65%) were male. Of the 43 cases, 22 (51%) experienced previous myelopathy and the other 21 (49%) were presenting for the first time.

### Heterogeneity in primary referrals of patients with DCM points to the lack of a unified pathway

Thirty-three primary referrals were accessible to review. The majority of these were made by a general practitioner (28, 85%) and less frequently by a physiotherapist (3, 9%) or other peripheral hospital (2, 6%). Half of all DCM cases were referred directly to a spine surgeon (22, 51%) while the other were seen by other specialties first (21, 49%), including neurology (n=16), orthopaedics (n=2), pain management (n=1), rheumatology (n=1) and geriatric medicine (n=1). Almost all cases (42, 98%) were ultimately reviewed by a spinal surgeon. Referral pathways are displayed separating new myelopathy cases from recurrent cases (figure 2).

### Patients with DCM encounter significant delays from the onset of symptoms to the time of primary referral

Symptom onset was defined as the point at which the patient first experienced symptoms. For example, if a primary referral stated the patient had been experiencing numbness in his fingers for 4 months. The onset of symptoms was defined as 4 months prior to the date of this referral. The times between symptom onset and primary referral for new and recurrent cases of DCM were 8.3±10.1 and 6.4±4.2 months, respectively. Seventy-six per cent (16/21) of new cases and 24% (5/22) of recurrent cases of DCM were referred to an alternative specialist for secondary assessment rather than directly to a spine surgeon. The times between primary referral and secondary assessment for new and recurrent cases of DCM were 1.5±0.8 and 1.9±1.2 months, respectively.

### Secondary referral to spine surgery is consistently delayed and does not take patient characteristics into account

Following secondary assessment, all patients apart from one were then referred on to see a spine surgeon. The one non-referred patient had a complex neurological and social background with mild myelopathy and was deemed better managed with regular neurology follow-up than consideration of spinal surgery. The times between secondary assessment and surgical assessment for new

**Table 1** Inferring modified Japanese Orthopaedic Association (i-mJOA) methodology

| | Score | mJOA | i-mJOA |
|---|---|---|---|
| Upper limb motor dysfunction | 0 | Inability to move hands | Power 0/5 or 1/5<br>No movement |
| | 1 | Inability to eat with a spoon but able to move hands | Power 2/5<br>Significant weakness/stiffness/clumsiness/impairment to daily living.<br>Significant assistance needed in daily living and/or totally dependent.<br>Can't hold objects. |
| | 2 | Inability to button shirt but able to eat with a spoon | Power 3/5<br>Marked weakness/stiffness/clumsiness/impairment to daily living.<br>Moderate assistance needed in daily living.<br>Frequently dropping objects. |
| | 3 | Able to button a shirt with great difficulty | Power 3/5<br>Moderate weakness/stiffness/clumsiness/impairment to daily living.<br>No/mild assistance needed in daily living.<br>Occasionally drops objects. |
| | 4 | Able to button a shirt with slight difficulty | Power 4/5<br>Mild weakness/stiffness/clumsiness/impairment to daily living |
| | 5 | No dysfunction | Power 5/5<br>Good strength<br>No weakness |
| Sphincter dysfunction | 0 | Inability to urinate voluntarily | Significant/Frequency urinary incontinence<br>Catheter required |
| | 1 | Marked difficulty with micturition | Moderate urinary symptoms<br>Moderate impairment to daily living<br>Any episode of urinary incontinence |
| | 2 | Mild to moderate difficulty with micturition | Mild urinary symptoms<br>Minor/no impairment to daily living<br>No episodes of incontinence |
| | 3 | Normal micturition | No urinary symptoms |

Continued

**Table 1** Continued

| | Score | mJOA | i-mJOA |
|---|---|---|---|
| Lower limb motor dysfunction | 0 | Complete loss of motor and sensory function | Power 0/5 or 1/5<br>Significant weakness without movement<br>Sensory loss |
| | 1 | Sensory preservation without ability to move legs | Power 0/5 or 1/5<br>Significant weakness without movement<br>Preserved sensation |
| | 2 | Able to move legs but unable to walk | Power 2/5<br>Significant weakness with minor movement<br>Difficulty in standing<br>Unable to walk |
| | 3 | Able to walk on flat floor with a walking aid (ie, cane or crutch) | Power 3/5<br>Marked weakness/stiffness/clumsiness/impairment to daily living<br>Walking aid |
| | 4 | Able to walk up and/or down stairs with a handrail | Power 3/5<br>Marked weakness/stiffness/clumsiness/impairment to daily living<br>No walking aid |
| | 5 | Moderate to significant lack of stability but able to walk up and/or downstairs without handrail | Power 4/5<br>Mild weakness/stiffness/clumsiness/impairment to daily living<br>Significant imbalance and/or falls |
| | 6 | Mild lack of stability but walk unaided with smooth reciprocation | Power 4/5<br>Mild weakness/stiffness/clumsiness/impairment to daily living<br>Some imbalance without falls |
| | 7 | No dysfunction | Power 5/5<br>Good strength<br>No weakness |
| Sensory dysfunction | 0 | Complete loss of hand sensation | Total loss of hand sensation |
| | 1 | Severe sensory loss or pain | Moderate/significant sensory loss<br>Pain more significant than numbness/paraesthesia |
| | 2 | Mild sensory loss | Mild/moderate sensory loss/paraesthesia |
| | 3 | No sensory loss | Normal sensation |

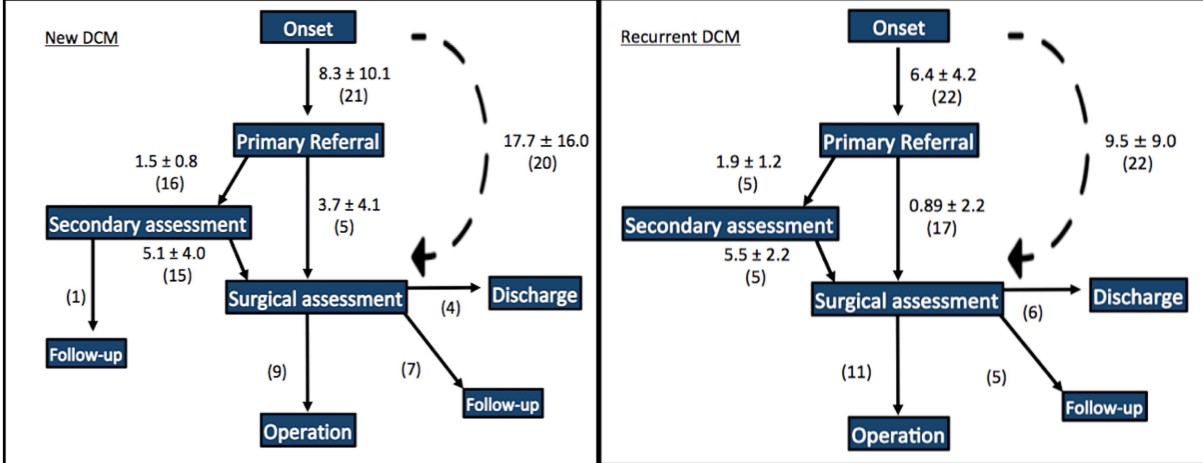

**Figure 2** Patient referral pathways from symptom onset for recurrent cases of DCM (left) and new cases of DCM (right). Times are given in the format: average months±SD (number of cases). Dashed lines represent total time durations, not a specific patient pathway. DCM, degenerative cervical myelopathy.

and recurrent cases of DCM were 5.1±4.0 and 5.5±2.2 months, respectively. No clinical feature, patient demographic factor or specialty of secondary assessment was associated with a significantly different speed of referral to spinal surgeon review.

### A minority of new patients with DCM are directly referred to spine surgery

Twenty-four per cent (5/21) of new cases and 77% (17/22) of recurrent cases were referred from primary assessment directly to a spine surgeon. The times between primary assessment and surgical assessment for these new and recurrent cases of DCM were 3.7±4.1 and 0.9±2.2 months, respectively. The only factor associated with direct referral to a spinal surgeon was previous DCM (p<0.001). The overall times between symptom onset and surgical assessment for new and recurrent cases of DCM were 17.7±16.0 and 9.5±9.0 months, respectively. Of the new cases of DCM, 45% (9/20) received an operation, 35% (7/20) received a follow-up appointment with a spine surgeon and 20% (4/20) were discharged. Of the recurrent cases of DCM, 50% (11/22) received an operation, 23% (5/22) received a follow-up appointment with a spine surgeon and 27% (6/22) were discharged.

### Significant delays in MRI scans relative to symptom onset may contribute to delayed diagnosis of DCM

The observed cohort was formed on the basis of diagnostic cervical MRI scans, demonstrating spinal cord compression. Relative to symptom onset, the scan was performed 17.7±15.6 and 9.5±8.7 months after symptomatic onset for new and recurrent cases of DCM, respectively. This scan was performed at 1.4±4.7 and 3.3±4.2 months after secondary assessment for new and recurrent cases of DCM, respectively. Relative to surgical review, it was performed at 2.3±0.7 months before and 0.5±1.2 months after surgical assessment for new and recurrent cases of DCM, respectively. Thus, in general, imaging of new DCM cases was conducted prior to reaching a spinal

surgeon, whereas recurrent cases were imaged after initial surgical consultation.

### Patients with DCM continue to deteriorate along the pathway from referral to treatment

Between the earliest recorded assessment (primary or secondary) and surgical assessment, the following deteriorations in i-mJOA scores were observed (figure 3). Fifty-nine per cent of patients deteriorated, 24% remained stable and 17% showed improvement, presumably due to symptom fluctuation or resolution of disc herniation. The total i-mJOA change for individual patients ranged from +4 to −6 points. On average, patients who deteriorated displayed a reduction of 2.3±1.8 i-mJOA points.

### DISCUSSION

The pathway from symptom onset to diagnosis and surgical assessment is complex and highly heterogeneous for patients with DCM. They face long delays before a diagnosis and further delays before surgical assessment. During this time, most experience a progression in symptoms. However, DCM symptoms are by at large, not reversible. The observed deterioration is, therefore, likely to translate into increased long-term disability and lead to lower quality of life.

### A diagnosis of DCM is made before reaching a spinal surgeon

Patients with DCM principally enter the healthcare system via primary care. They are then referred via another specialty for assessment and imaging, typically neurology, before reaching a spinal surgeon. This contrasts the study by Behrbalk *et al*, the only other study to consider the DCM healthcare system, which found most patients were initially assessed by orthopaedics.[18] A potential explanation is that patients may self-refer to orthopaedics in Israel but require a formal referral to neurology or neurosurgery. Regardless, both studies have demonstrated that

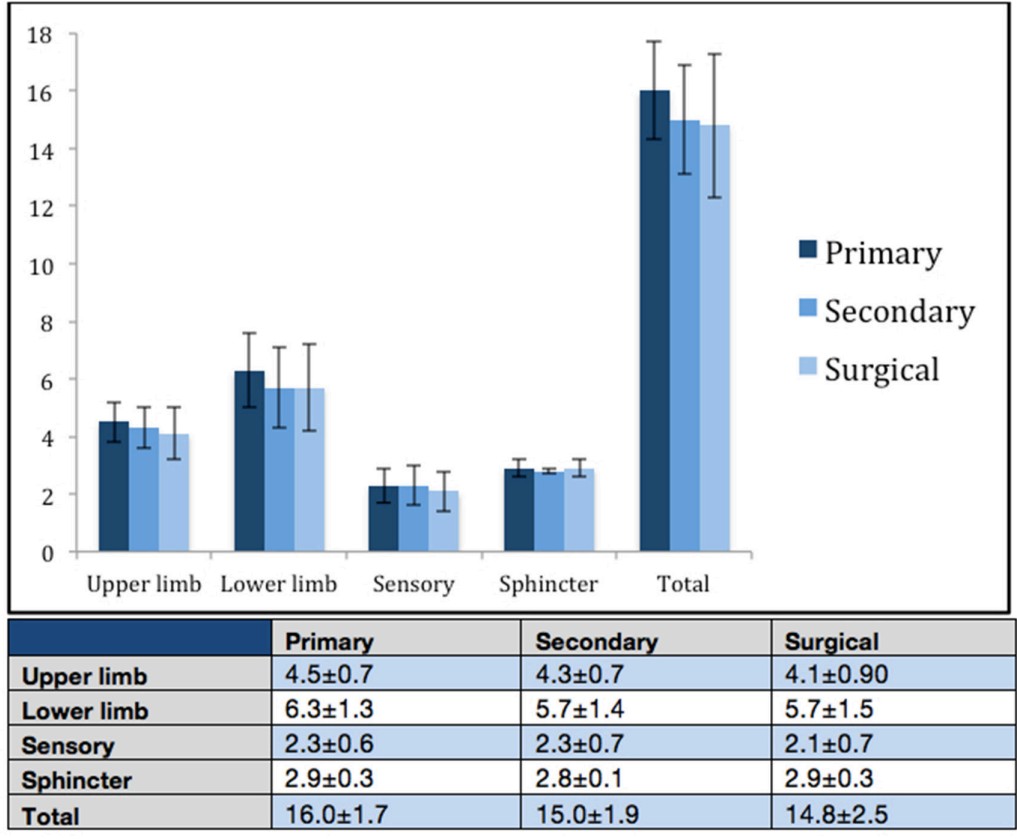

| | Primary | Secondary | Surgical |
|---|---|---|---|
| **Upper limb** | 4.5±0.7 | 4.3±0.7 | 4.1±0.90 |
| **Lower limb** | 6.3±1.3 | 5.7±1.4 | 5.7±1.5 |
| **Sensory** | 2.3±0.6 | 2.3±0.7 | 2.1±0.7 |
| **Sphincter** | 2.9±0.3 | 2.8±0.1 | 2.9±0.3 |
| **Total** | 16.0±1.7 | 15.0±1.9 | 14.8±2.5 |

**Figure 3** i-mJOA scores at each clinical phase of assessment. Mean±SD. i-mJOA, Inferring modified Japanese Orthopaedic Association.

the diagnosis of new-onset DCM is made outside of spinal surgery.

This finding is significant as DCM treatment and research is largely led by spinal surgeons. In order to accelerate diagnosis and improve outcomes based on existing treatment paradigms, greater clarity of the interaction between other specialties and patients with DCM is critical.

### Patients with DCM wait months for a diagnosis and years for treatment

Delays in patient pathways were noted in all components of the system. The greatest delay appears to exist between disease onset and primary referral to secondary care. This is consistent with Behrbalk *et al* and patient experience.[18]

The cause for this delay is likely to be multifactorial. Potential factors include late presentation by patients, subtle or non-specific symptoms and examination findings,[24 25] misdiagnoses and a lack of awareness of DCM. Future studies are required to determine the exact reasons for prereferral delays in order to develop targeted interventions.

Minimising delays of treatment is especially important in DCM, because length of symptoms and disease severity are the two most important factors that determine response to surgery of patients with DCM.[13–15] Moreover, shortening the times until patients are seen by a myelopathy specialist is likely to be the single most effective strategy of improving patient outcomes.

### The lack of a unified pathway is likely to contribute to delays in treatment

The majority of new cases (76%) were not directly referred to a spine surgeon but instead were reviewed by neurologists, orthopaedics, pain specialists, rheumatologists and geriatricians. The specialty performing the secondary assessment did not influence referral speed from secondary care to spinal surgery. These findings corroborate that there is no streamlined pathway for surgical assessment of a patient with DCM.

At the early stages, symptoms are often subtle and often missed by non-specialists. However, even asymptomatic patients with MRI-proven cervical spinal cord compression run a considerable risk of developing myelopathy. A recent study found that 59% of asymptomatic individuals over 40 had cervical cord compression on MRI.[26] A further study observed that 13% of non-myelopathic individuals with cervical cord compression developed myelopathy within a 2-year period.[27]

Our data indicated that disease severity did not influence speed of referral to a spine surgeon. This may seem counterintuitive but potentially be explained by the broad range of symptoms associated with myelopathy.

For these reasons, the authors advocate more rapid assessment by a specialist service once a patient has proven cord compression, especially alongside clinically myelopathic features. This is also reflected in the first recent international guidelines.[10]

### Patients with DCM continue to deteriorate while they wait for treatment

This study found that 59% of patients deteriorated between their earliest available assessment and the point at which they reached assessment by a spine surgeon. The minimum clinically important difference in DCM varies by disease severity[28 29]: 1 point for mild disease (mJOA ≥15), 2 points for moderate disease (mJOA score of 12–14) and 3 points for severe disease (mJOA score <12). In the deteriorating group, all but one patient were found to have a mild disease (mJOA ≥15) at initial assessment. Within this group, the average deterioration was 2.3 i-mJOA points, demonstrating clinically significant disease progression. More rapid DCM referral services and streamlined pathways to specialist assessment could help to prevent this source of potentially non-reversible deterioration.

### What would the ideal pathway be for assessment and management of DCM?

As it stands, these data show new diagnosis and management as a two-stage process. Stage 1: patients receive a diagnosis of DCM outside of spinal surgery based on clinical symptomatology and MRI findings. Stage 2: onward referral to spinal surgery and surgery is offered if indicated.

While direct referral to spinal surgery on initial clinical suspicion would significantly shorten this process, it would likely lead to a number of problems. First, given the nature of DCM's non-specific symptoms, especially early on, and the limited access in the UK to MRI imaging within primary care, this would lead to many inappropriate referrals and place a strain on already stressed outpatient spinal services. Second, these inappropriate referrals are then being assessed by specialists, who can confidently consider DCM, but will not be best placed or experienced to consider other differentials. Therefore, the exclusion of myelopathy could place delays on other diagnoses.

Similar dilemmas have been addressed for many other conditions using intermediate pathways. For example, in the UK, we have a 'musculoskeletal' pathway triaging complaints such as lower back pain and joint pain. In this pathway, patients are referred to physiotherapy-led assessment centres, with access to investigations such as MRI and electrophysiology. These centres have access to specialist multidisciplinary teams (MDTs), who help to decide whether patients can be managed locally or referred to specialist services. A recent systematic review has shown such triage was concordant with specialists 68%–96% of the time and reduced referrals to specialist clinics by 20%–60%.[30]

At present, the 'musculoskeletal' service is better geared to classically 'orthopaedic' or 'rheumatological' conditions and a potential weakness for DCM would be neurological mimics. In this study, only 9% of referrals came from this musculoskeletal service, with most patients being referred to neurology. A more refined assessment process will require further research. Building on the existing intermediate pathway with closer involvement of neurology might be a sensible approach.

For recurrent disease, this study shows that direct access to spinal surgery is appropriate. Of the patients referred, 50% received operations and 23% were kept under surveillance. We speculate that these patients' previous DCM diagnosis heightened awareness by patient and healthcare practitioner, leading to a greater confidence in direct referral to spinal services.

### Limitations

While this study was conducted in a single neurosciences centre, it serves a large regional population, almost a tenth of the UK population as a whole.[31] Additionally, given the UK has a National Health Service that delivers care within common structures, the findings in this study are likely generalisable throughout at least the UK health system.

The epidemiology of DCM is poorly characterised, however, it is thought to be a common condition.[32] The identification of only 43 cases of DCM over 1 year was lower than expected. However, the majority of patients receiving treatment at this neurosciences centre present with external imaging obtained in referring secondary centres, which did not facilitate access to full clinical records as described in the methodology. Hence, these patients were not included in the present cohort. The low number, therefore, is likely to reflect the role of the neuroradiology department acting as a secondary centre serving a more local population.

The retrospective nature of the study leads to the development of i-mJOA. While true mJOA would have been preferable, the strong kappa statistics and low average difference indicate that total i-mJOA does represent a reliable proxy and broad-brush method of demonstrating disease severity and changes in disease severity over time retrospectively.

### CONCLUSIONS

This work demonstrates that patients experience significant delays in diagnosis and time to surgical review. During this time, disease progression occurred and patients functionally deteriorated. The best postoperative outcomes for patients with DCM are attained with rapid surgical review and intervention when appropriate. Our findings call for the formation of dedicated streamlined pathways to surgical review for patients with spinal cord compression on MRI, especially in the context of clinical myelopathy. Further prospective system characterisation of factors associated with patient flow is needed to

introduce specific targeted interventions to improve the management of DCM in the UK.

**Acknowledgements** Research in the senior author's laboratory is supported by a core support grant from the Wellcome Trust and MRC to the Wellcome Trust-Medical Research Council Cambridge Stem Cell Institute. MRNK is supported by an NIHR Clinician Scientist Award.

**Contributors** BH: study design, data collection, data analysis, manuscript writing. JT-M: data collection, manuscript editing. BD: study design, manuscript editing. MK: study design, manuscript editing.

**Funding** The authors have not declared a specific grant for this research from any funding agency in the public, commercial or not-for-profit sectors.

**Disclaimer** This report is independent research arising from a Clinician Scientist Award, CS- 2015-15-023, supported by the National Institute for Health Research. The views expressed in this publication are those of the authors and not necessarily those of the NHS, the National Institute for Health Research or the Department of Health and Social Care.

**Competing interests** None declared.

**Ethics approval** All data were collected as part of a service evaluation and appropriate prior permission was sought and granted.

**Provenance and peer review** Not commissioned; externally peer reviewed.

**Data sharing statement** No additional data are available.

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
