## [Reviewer comments · BMJ Open]

ARTICLE DETAILS

TITLE (PROVISIONAL)	Route to diagnosis of degenerative cervical myelopathy in a UK healthcare system: a retrospective cohort study
AUTHORS	Hilton, Bryn; Tempest-Mitchell, Jennifer; Davies, Benjamin; Kotter, Mark

VERSION 1 – REVIEW

REVIEWER	Andrei F Joaquim University of Campinas - Campinas-SP, Brazil
REVIEW RETURNED	22-Apr-2018

GENERAL COMMENTS	the paper is well written and discuss a common problem: delay in treatment of Cervical degenerative disease I would recommend to evaluate and present more data about delay in CDD already reported in literature. Add information in the discussion about similar papers who evaluated the delays in treatment are extremely necessary for comparison
---

REVIEWER	Paul C. Willems, MD, PhD Dept of Orthopedics, Maastricht University Medical Centre, The Netherlands
REVIEW RETURNED	23-Jun-2018

GENERAL COMMENTS	This study was conducted in a single tertiary Neurosciences Centre in which the radiological reports of all cervical spine MRI scans conducted over one year were retrospectively screened for cervical cord compression. Then, if positive, corresponding clinical records were screened for degenerative cervical myelopathy. If so, it was studied what had been the time between onset of symptoms and eventual treatment (surgery) and whether in that time period there had been a deterioration in neurological status as assessed by a self-modified Japanese Orthopedic Association Score. Methodologically, I have two concerns: -In this academic center many patients came from hospitals from another region who had made their MRI elsewhere, so these patients were not included in this study which constitutes a selection bias, limiting generalization of the findings as only partially acknowledged by the authors in Limitations section-The authors used a self modified version of the JOA score for neurological status. This so-called inferred version was retrospectively based on reports from physicians/family doctors which makes it highly susceptible for bias. Additionally, the inferred method itself has never been published, and in exactly one sentence in the Methods section the results of a prospective validation the authors executed themselves, is reported. I think this
---

	validation should be reported much more in detail, preferably performed by others, and it should be acknowledged much more clearly that this is a serious limitation of the study. Overall, I think this is an interesting study which shows that there are currently significant delays in diagnosis and time to surgical review for patients with DCM, during which disease progression may occur. This is an important finding which suggests that higher awareness in primary care is needed for earlier referral and more directly to the proper medical specialist (surgeon). This suggests that another organisation/referral pattern, especially in first line health care is advocated. This could preferably best be supplied by an integrated multidisciplinary supervised spine setting in first line care, which is more easily accessible for patients and where by adequate triage and close communication with medical specialists) direct referral to the appropriate medical specialist can be arranged in due time. In this way, the results of the current study may contribute to a better organization of care not only for DCM patients but for all (spine) patients in general.
--	---

REVIEWER	Catherine Jutzeler University of British Columbia
REVIEW RETURNED	25-Jun-2018

GENERAL COMMENTS	Hilton et al performed a retrospective observational study to characterize the DCM patient pathways within the UK healthcare system in order to support targeted interventions aimed at accelerating diagnosis and management. This is a well-designed and well-written study. There are some comments to be addressed:  1) The introduction is very lengthy and could be shorten (e.g., part about epidemiology and symptoms) 2) The authors should state the design of their epidemiological study (retrospective and observational) at the beginning of the method section. 3) If the study was registered at clinicaltrial.gov, please provide the NTC identifier. 4) Please specify who screened the radiological reports. 5) How was DCM defined? Was the mJOA a part of the assessment? If not, please explain. 6) As the onset of DCM occurs often before the patient presents to a physician, I am wondering how the authors define the age at time of onset? Or do the authors refer to the age the patient was diagnosed? In that case, it might be better to talk about age of diagnosis? 7) In the results section and figure 1, the authors state: '1123 scans were screened, resulting in a final cohort of 43 DCM cases'. Were these scans of 1123 patients? Or 1123 scans of which some are from the same patient? Please clarify this as this might be not clear to readers who are not experts in the field. 8) In the chart flow (figure 1) it is indicated that 85 patients with a spinal cord compression were excluded due to no DCM. Did the authors follow up on these patients to assess how many of these patients have developed a DCM later? 9) The authors should also fill in the STROBE checklist and submit with the manuscript Table
---

	The authors are encouraged to provide a table with individual data: demographics, i-mJOA scores at each clinical phase assessment (i.e., Figure 3 should then be removed), radiological findings, initials symptoms, clinical and radiological evidence of the deterioration whilst patients wait for treatment. This information would be very helpful for the reader to interpret the data and results. Figure 2: It is not clear what the difference between 'regular follow-up' and 'follow-up' is. Please clarify. Please explain in the figure legend what the dashed lines mean. Is there a reason why 'regular follow-up' is not in a blue box? Please define onset. From this figure it is not clear if the onset of DCM is patient reported (i.e., when the patient first noticed symptoms) or refers to the first time the patient was seen by a physician or physical therapist? Who performed the secondary assessment? Spine surgeons? Or other specialists? The figure quality (it is blurry and difficult to interpret) appears to be low. This could be due to the PDF version this reviewer has access to. But it might be worth it to double check.
--	--

VERSION 1 – AUTHOR RESPONSE

Reviewer 1

Reviewer 1 felt that **“the paper is well written and discuss[es] a common problem”**. Reviewer 1’s only suggested point of revision was to “evaluate and present more data about delay in cervical degenerative disease already reported in the literature” and to add “similar papers who evaluated the delays in treatment”.

To the authors’ knowledge, we have presented the extent of literature surrounding diagnostic delays in DCM. The single paper that has been previously published focusing on this issue was Behrbalk et al 2013, which we cover in our Discussion section. It is worth noting that these authors cited *“they have found no published data on the process and time frame involved in attaining a diagnosis of CSM”*. There has certainly been nothing published since.

The reviewer’s comment likely reflects that delays in diagnosis are well recognised anecdotally amongst spinal surgeons and patients. Feedback and discussion within two conferences in the UK and one in Europe at which this research was presented confirmed that diagnostic delays are a well-recognised but almost entirely unreported phenomenon.

Understanding and improving diagnostic delays are a significant priority for patients; this was noted by a forum of patients from Myelopathy.org (“Elemental Ideas”) and lead to a recent Easily Missed article in the BMJ (Davies et al 2018).

Therefore, the authors feel this work would make a substantial contribution to the literature concerning management of DCM.

Behrbalk E, Salame K, Regev GJ, Keynan O, Boszczyk B, Lidar Z. Delayed diagnosis of cervical spondylotic myelopathy by primary care physicians. *Neurosurg Focus*. 2013 Jul;35(1):E1

“Elemental Ideas - Cervical Spondylotic Myelopathy.” Cambridge TV. www.myelopathy.org, www.cambridge-tv.co.uk/tag/myelopathy/.

Davies BM, Mowforth OD, Smith EK, Kotter MR. Degenerative cervical myelopathy. *BMJ*. 2018 Feb 22;360:k186

Conferences:

Joint meeting between the German Society of Neurosurgery and the Society of British Neurological Surgeons 2017 - Madgeburg, Germany.

Society of British Neurological Surgeons Autumn Conference 2017 – Liverpool, UK.

Society of British Neurological Surgeons Spring Conference 2016 – Oxford, UK.

Reviewer 2

Reviewer 2 describes the authors’ work as an **“Interesting study... [which presents] an important finding which suggests that higher awareness in primary care is needed for earlier referral and more directly to the proper medical specialist (surgeon).”** Reviewer 2 also goes on to propose a potential solution for the reported diagnostic delays: using an **“integrated multidisciplinary supervised spine setting in first line care...the results of the current study may contribute to a better organization of care not only for DCM patients but for all (spine) patients in general”**. We are very grateful for this feedback and their proposed solution.

Reviewer 2 raises two methodological issues with our manuscript.

Issue 1:

Reviewer 2 highlights a “selection bias” introduced by the fact that the authors chose to exclude patients who had their MRI scan conducted in peripheral hospitals and were subsequently referred on to our tertiary neurosciences centre for further specialist management. The reviewer did not feel that this limitation was sufficiently addressed in our manuscript. This is a helpful observation and we appreciate the chance to clarify our methodology.

The goal of this research was to track patients from onset of symptomatic myelopathy to the point at which they were assessed by a spinal surgeon for consideration of surgery. Peripheral hospitals in our region use different electronic and paper record keeping systems from one another and from our centre. Therefore, inclusion of peripherally referred patients would leave us only with their MRI scan imaging and report and the clinical record of their assessment by one of the spine surgeons working in our centre. This would not provide sufficient data to track their disease course and referral pathway.

As well as acting as a tertiary referral centre, our hospital also acts as a large regional hospital serving a diverse and sizeable local population. Therefore, inclusion of only patients imaged at our centre facilitated access to all records of these patients and proper documentation of their disease journeys.

The authors will gladly amend the methods section to clarify this point whilst acknowledging the specific limitation this brings to generalisation of our results to peripherally referred patients.

Issue 2:

Reviewer 2 raises concern over the use of our inferred mJOA (i-mJOA) scoring system. Reviewer 2 states, i-mJOA is “retrospectively based on reports from physicians/family doctors which makes it highly susceptible for bias”, “I think this validation should be reported much more in detail, preferably performed by others, and it should be acknowledged much more clearly that this is a serious limitation of the study.”

The authors fully acknowledge the limitation of the i-mJOA scoring and will further elaborate on its development and limitations.

Rationale for Development / Intended Use

It was developed as a research tool for this study to allow retrospective assessment of change in DCM disability, as grading tools are not routinely employed in clinical consultations. It was not specifically developed for wider usage (which would clearly require external validation). The authors wished to demonstrate that neurological and functional deterioration occurs over the delayed time to diagnosis, not that an absolute value or rate for deterioration could be determined.

Development Methodology

The authors appreciate that our methodology for design of i-mJOA was not fully described in our manuscript. We are of course happy to provide further details:

1. Author BH designed a translation of mJOA into i-mJOA based upon commonly reported clinical details in patients' electronic records (e.g. “walking aid” vs. “no walking aid” imbalance, or 0-5/5 power grading). This translation is presented in Table 1. Other similar translations of the mJOA

score have previously been conducted. Most recently the P-mJOA was created for ease of patient use and was validated against true mJOA scores with similar results to our i-mJOA (Rhee et al.

2018).

2. The electronic records of 13 patients assessed by a spinal clinic in our centre were collected by author JTM. These were patients outside of the study range dates and thus were not included in our study. mJOA assessments were made during their consultation and therefore a formal mJOA score was available.
3. These records were then anonymised by JTM of any identifying details and the mJOA questionnaire results were removed.
4. Author BH then applied the mJOA to i-mJOA translation to these patient records.
5. True mJOA and i-mJOA scores were then statistically analysed for validation and these results are presented in the manuscript.
6. Total mJOA and total i-mJOA scores showed very close correlation, 0.13 ± 1.51 (kappa 0.73). Thus, the authors felt that the i-mJOA was a suitable tool to apply for this study in order to capture the neurological deterioration that was evident in patients clinical records.

Once again, i-mJOA was not intended to be a perfectly accurate tool for retrospectively creating mJOA scores, but rather a broad-brush method of demonstrating disease severity and changes in disease severity over time retrospectively. Therefore, whilst acknowledging its clear limitations, the authors do not feel i-mJOA constitutes a methodological problem.

Rhee JM, Shi WJ, Cyriac M et al. The P-mJOA: A Patient-derived, Self-reported Outcome Instrument for Evaluating Cervical Myelopathy: Comparison with the mJOA. Clin Spine Surg.

2018 Mar;31(2):E115-E120.

Reviewer 3

Reviewer 3 states that this is a “**well-designed and well-written study.**” Reviewer 3 very helpfully lists a number of specific points that require further clarification. Each point is individually addressed below.

Text:

- 1) *The introduction is very lengthy and could be shorten (e.g., part about epidemiology and symptoms)*

The authors agree that the introduction is detailed. The purpose of our lengthy introduction to DCM was that we hope this paper will reach an audience extending beyond the spinal community (e.g.

primary care) who may need further background on the topic of DCM especially in terms of its epidemiology. We are of course happy to consolidate wherever possible without losing any of the key information for those readers less familiar with DCM.

- 2) *The authors should state the design of their epidemiological study (retrospective and observational) at the beginning of the method section.*

The authors are happy to make this addition.

- 3) *If the study was registered at clinicaltrials.gov, please provide the NTC identifier.*

The study was not registered at clinicaltrials.gov. The data was collected as part of a broad service evaluation project.

- 4) *Please specify who screened the radiological reports.*

The radiological reports were all read and screened by author BH.

- 5) *How was DCM defined? Was the mJOA a part of the assessment? If not, please explain.*

DCM was defined as spinal cord compression on MRI and the presence of a clinical diagnosis of myelopathy in the patient's clinical records.

mJOA was unfortunately not part of most assessments as it is not routinely calculated during consultations, especially across specialities. Thus, we designed the i-mJOA to serve by proxy in its place as a gross marker for neurological disability and disease severity.

- 6) *As the onset of DCM occurs often before the patient presents to a physician, I am wondering how the authors define the age at time of onset? Or do the authors refer to the age the patient was diagnosed? In that case, it might be better to talk about age of diagnosis?*

The time of onset was calculated by examining the referral letter from primary care to secondary care. Each letter stated a time frame for symptoms. E.g. Mr X has been suffering from imbalance and difficulties with his dexterity for 3 months". 3 months prior to the date of this referral was taken to be the onset of symptoms. The authors will further clarify this point in the manuscript.

- 7) *In the results section and figure 1, the authors state: '1123 scans were screened, resulting in a final cohort of 43 DCM cases'. Were these scans of 1123 patients? Or 1123 scans of which some are from the same patient? Please clarify this as this might be not clear to readers who are not experts in the field.*

1123 was the number of scan reports reviewed. Rarely patients were re-imaged within the study time frame. The authors will amend this for clarification.

8) *In the chart flow (figure 1) it is indicated that 85 patients with cervical spinal cord compression were excluded due to no DCM. Did the authors follow up on these patients to assess how many of these patients have developed a DCM later?*

No, the authors did not follow these patients up. A small number of these would be expected to become myelopathic within the time duration of this study. Furthermore, given that the goal of the study was to examine patients from symptom onset to spinal surgeon review, asymptomatic patients with spinal cord compression would most likely experience different diagnostic pathways and would be a distinct but important project in its own right to follow these patients up over time.

9) *The authors should also fill in the STROBE checklist and submit with the manuscript* The authors will gladly provide a STROBE checklist as part of the submission of a revised manuscript.

Table:

The authors are encouraged to provide a table with individual data: demographics, i-mJOA scores at each clinical phase assessment (i.e., Figure 3 should then be removed), radiological findings, initials symptoms, clinical and radiological evidence of the deterioration whilst patients wait for treatment. This information would be very helpful for the reader to interpret the data and results.

The authors did originally consider producing such a table but given the sheer volume of information such a table would need to include to adequately depict the cases of 43 patients, we felt this would require a relatively large and inaccessible table.

Figure 2:

1) *It is not clear what the difference between 'regular follow-up' and 'follow-up' is. Please clarify.*

Thank you for pointing this out. It was written in error. 'regular follow-up' simply referred to follow-up outside of spinal surgical services (e.g. by a neurologist in this case). The authors will amend the figure to clarify.

2) *Please explain in the figure legend what the dashed lines mean.*

The dashed lines depict total time durations for ease of visualisation, not a specific patient pathway. The authors will clarify this in the figure legend.

3) *Is there a reason why 'regular follow-up' is not in a blue box?*

No, the authors will correct this and clarify as described above.

4) *Please define onset. From this figure it is not clear if the onset of DCM is patient reported (i.e., when the patient first noticed symptoms) or refers to the first time the patient was seen by a physician or physical therapist?*

Onset refers to the patient-experienced symptom onset. The authors will clarify this in the image.

5) *Who performed the secondary assessment? Spine surgeons? Or other specialists?*

Secondary assessment was performed by a range of non-spine surgeon specialists as described in the main body of the manuscript: neurology (N=16), orthopaedics (N=2; nonspinal orthopaedics), pain management (N=1), rheumatology (N=1) and geriatric medicine (N=1).

6) *The figure quality (it is blurry and difficult to interpret) appears to be low. This could be due to the PDF version this reviewer has access to. But it might be worth it to double check.*

We apologise for this issue. The image meets the specifications set by the journal. Perhaps an issue has arisen with the PDF version. We are happy to liaise with the editorial assistant to ensure that the quality is sufficient for further review.

VERSION 2 – REVIEW

REVIEWER	Andrei F Joaquim Professor of Neurosurgery, University of Campinas, Brazil
REVIEW RETURNED	10-Oct-2018
GENERAL COMMENTS	Thank you for the opportunity to review this paper. In my first review, I already believe that the manuscript was able to be accepted. after some additional revisions, I don t have any other comment, besides to congratulate the authors. thank you very much
REVIEWER	Catherine Jutzeler University of British Columbia, Vancouver, Canada
REVIEW RETURNED	15-Oct-2018

GENERAL COMMENTS	Revisions to the previous manuscript were thoughtfully done. Thank you.
--

VERSION 2 – AUTHOR RESPONSE

Thank you for your correspondence regarding this manuscript. We were pleased to read the favorable reviews from the two reviewers. I have made changes to the manuscript to address the points that you raised regarding formatting issues. I trust everything should now be correct but please do let me know if there are any further issues and I will remedy them as swiftly as possible.